# Role of Essential Amino Acids in Age-Induced Bone Loss

**DOI:** 10.3390/ijms231911281

**Published:** 2022-09-24

**Authors:** Ziquan Lv, Wenbiao Shi, Qian Zhang

**Affiliations:** 1Department of Nutrition and Health, China Agricultural University, Beijing 100193, China; 2Shenzhen Center for Disease Control and Prevention, Shenzhen 518055, China

**Keywords:** essential amino acids, bone mass, bone mineral density, aging

## Abstract

Age-induced osteoporosis is a global problem. Essential amino acids (EAAs) work as an energy source and a molecular pathway modulator in bone, but their functions have not been systematically reviewed in aging bone. This study aimed to discuss the contribution of EAAs on aging bone from in vitro, in vivo, and human investigations. In aged people with osteoporosis, serum EAAs were detected changing up and down, without a well-established conclusion. The supply of EAAs in aged people either rescued or did not affect bone mineral density (BMD) and bone volume. In most signaling studies, EAAs were proven to increase bone mass. Lysine, threonine, methionine, tryptophan, and isoleucine can increase osteoblast proliferation, activation, and differentiation, and decrease osteoclast activity. Oxidized L-tryptophan promotes bone marrow stem cells (BMSCs) differentiating into osteoblasts. However, the oxidation product of tryptophan called kynurenine increases osteoclast activity, and enhances the differentiation of adipocytes from BMSCs. Taken together, in terms of bone minerals and volume, more views consider EAAs to have a positive effect on aging bone, but the function of EAAs in bone metabolism has not been fully demonstrated and more studies are needed in this area in the future.

## 1. Introduction

The aging process is characterized by changes in body composition, such as an increase in body fat, and a reduction in lean mass (sarcopenia) and bone density (osteopenia) [1]. Osteopenia happens as a consequence of imbalanced osteoblast and osteoclast activity. Osteoblasts are differentiated from BMSCs and are responsible for secreting the bone matrix, which will be mineralized in bone development and homeostasis [2]. An osteoclast is the original bone-resorbing cell from monocytes/macrophages [3]. Osteopenia increases the incidence of bone fractures, is one of the largest musculoskeletal diseases, and is a main reason for the death of the elderly. In people older than 65, women have an absolute mortality risk of 12.5% within 1-year post-fracture, while men have a risk of 19.5% [4].

Normally, bone is considered as a supporting structure, but actually, it is also a very active metabolic organ. In addition to mineral metabolism, as the main energy source, glucose, fatty acid, and amino acid metabolism are also studied in bone. Glucose and fatty acids both work as energy sources and are bone structure providers; additionally, they play major roles in inducing internal signaling pathways in bone. For instance, glucose can induce the mammalian target of the rapamycin (mTOR) pathway in osteoblasts [5], which promotes osteoblast differentiation from pre-osteoblasts [6]. Fatty acids function as a critical source of adenosine triphosphate (ATP) in osteoblasts and osteocytes. Intriguingly, disrupting the carnitine palmitoyltransferase 2 (Cpt2)-regulated fatty acid uptake pathway in bone impaired the postnatal bone acquisition in female mice, but not in male mice [7].

Twenty-five years ago, some researchers thought that excess dietary protein would induce more bone resorption by causing extra serum potassium and excreting more acid [8]. However, this view has been challenged by more recent studies. Dietary proteins are recommended to improve bone loss by forming an organic matrix of bone such as collagen, as indicated by a cohort study from 1966 to 2008. This study showed a diet containing 25–30% of total energy from protein every day, and 1.4 g protein/kg/day with three daily servings of dairy had a positive effect on lumbar spine BMD in adult people [9].

Proteins are formed by amino acids. There are eight EAAs: lysine (Lys), tryptophan (Trp), phenylalanine (Phe), methionine (Met), threonine (Thr), isoleucine (Ile), leucine (Leu), and valine (Val). Branched-chain amino acids (BCAAs) are amino acids having an aliphatic side chain with a branch. In EAAs, isoleucine, leucine, and valine are BCAAs [10], which are more intensively studied, mostly because previous studies found that they are effective in losing weight and keeping muscle. People have to absorb EAAs from food. According to the Cleveland Clinic, the best sources of EAAs are animal proteins, as they are the most easily absorbed and used by the human body. Complete proteins, including beef, poultry, fish, eggs, dairy, soy, quinoa, and buckwheat, contain all the EAAs. Incomplete proteins, such as nuts, some grains, and vegetables, mainly contain one or several EAAs [11] (Figure 1). Leucine, isoleucine, valine, serine, and threonine have been reported to correlate with an increase in longevity [12]. Muscle and bone are two interconnected tissues [13]. EAA supplementation has been shown to improve muscle function in the elderly with normal activity [14,15] and during bed rest [16]. However, the effect on age-related osteoporosis was not reported in this study. The functions of EAAs in aging bone have not been systematically reviewed. Since there is more and more knowledge about bone as a metabolic organ, EAAs should be studied more with regard to aging bone in the future. This study aimed to discuss the contribution of EAAs in aging bone as an energy provider and as a molecular signal modulator both in vitro and in vivo.

## 2. Amino Acid as Energy Source and Tissue Builder in Bones

Citrate is an intermediate product in the tricarboxylic acid cycle (TCA cycle, Krebs cycle), which means it can be generated by the metabolism of glucose, and fatty and amino acids. Dickens et al. proved that approximately 90% of citrate was in mineralized tissue in mammals [17]. In the recent 10 years, with solid-state nuclear magnetic resonance spectroscopy (NMR) and other higher technologies, citrate has been found incorporated in the calcium–phosphate structure, to form crystals in mineralized bone, which is the foundation of hard tissue [18,19,20]. The citrate content levels of bone and plasma are both reduced in osteoporotic and aged animals [21]. Therefore, the source of mineral citrate has been an interesting topic in the recent few years. In 2018, X Fu et al. proved that part of the citrate is directly generated from glucose from BMSCs upon differentiation into osteoblasts [22].

The mass spectrometry technique is used to detect metabolic tracking from stable isotope tracers [23]. 13C is the predominant isotope used because of its universal distribution in almost every cellular organic molecule [24]. In metabolic substrate analysis, M + 0 is an unlabeled molecule, while M + 1, M + 2, M + 3, etc., are different numbers of carbon labeled with 13C [25]. Using 13C-tracing technology, Y Yu et al. demonstrated that in differentiated mineralized skeletal stem cells (SSCs), glutamine’s contribution to citrate (M + 4 and M + 5) was reduced compared with undifferentiated SSCs, which means glutamine contributes more during the differentiation procedure, but is not the major energetic substrate in osteoblasts [26]. Although there is no direct evidence of citrate produced by EAAs, from Yu’s study, we can hypothesize that EAAs also might not be the major energy source in osteoblasts, while whether they are the source of citrate during differentiation from SSCs to osteoblasts still needs to be investigated.

## 3. Amino Acid Sensing and Bone Metabolism

The ability to sense and react to amino acids is important in bone as well as other tissues. General control nonderepressible 2 (GCN2) is an important amino acid sensor triggered by amino acid deprivation. In drosophila, GCN2 is associated with diet restriction-induced life span extension [27]. The Eukaryotic Translation Initiation Factor 2 Alpha Kinase 4 (Eif2ak4)/GCN2/Activating Transcription Factor 4 (ATF4) pathway is required for the uptake of amino acids and proliferation of cells in primary BMSCs [28]. In osteoblasts, the GCN2/ATF4 pathway is also needed for osteoblast differentiation and bone formation in response to wingless and INT (WNT) signaling, but Eif2ak4 is not required in osteoblasts differentiation [29] (Figure 2A).

Another key amino acid sensor is mTOR, which is decreased in calorie restriction-induced longevity [30]. Both rapamycin treatment [31] and the genetic deletion of mTOR downstream p70 ribosomal protein S6 kinase 1 (S6K1) [32] can extend lifespan. There are two protein complexes of mTOR, mTORC1 and mTORC2. mTORC1 can be stimulated by amino acids while mTORC2 is not sensitive to amino acid sensing [33]. The inhibition of mTORC1 with rapamycin impairs BMSC proliferation and differentiation, inducing trabecular bone loss [34]. WNT promotes mTORC1 signaling to mediate glutamine catabolism, resulting in increased osteoblast anabolism [29]. Moreover, bone morphogenetic proteins (BMP) also induce osteoblast differentiation via mTORC1 signaling in vivo [35]. These studies elucidate that in young mice, mTORC1 works as an anabolic signal for bone formation, mainly through increasing BMSC and osteoblast proliferation and differentiation (Figure 2A). In 2021, L Stukenforg et al. found that in BMSCs isolated from older individuals characterized by cell senescence, the inhibition of the mTOR pathway could prevent senescence, extend BMSC proliferation, and maintain stemness of the cells [36]. In an accelerated aging disorder mouse model with Hutchinson–Gilford progeria syndrome (HGPS), H Liu et al. showed that mTORC1 signaling was increased in stress-induced MSC senescence, and also in a D-galactose-induced osteoporosis model in rats. The inhibition of the mTORC1 pathway rescued the senescence phenotype in vitro [37] (Figure 2B). Taken together, the GCN2 signal has not been investigated in aging bone yet, while the role of mTOR in aged-induced osteoporosis models remains controversial, especially in vivo. There is a lot to be done to study the role of mTOR and GCN2 signaling in an aging bone model.

## 4. The Effect of EAAs on Bone Health in Animals and Cells

Studies with mouse models and cells have mostly focused on the mechanism underlying how EAAs regulate bone health. In addition to population-based surveys, there are also studies regarding EAAs in primary cells isolated from human bone. Residing in the bone marrow niche, BMSCs are multipotent stem cells able to differentiate into osteoblasts, adipocytes, and chondroblasts [38]. In human primary osteoblasts derived from both healthy bone and osteopenic bone, the supplementation of lysine has a positive effect on osteoblast proliferation, activation, and differentiation by inducing nitric oxide (NO), insulin-like growth factor-I (IGF1), osteocalcin, and alkaline phosphatase (ALP) activities [39]. In cultured human osteoblasts, the administration of lysine, but not arginine, reduced levels of transforming growth factor-β1 (TGF-β1) [40], which is produced by osteoblast-like cells and mediates normal bone remodeling. In addition to TGF-β1, lysine exposure also inhibited IL-6 levels, suggesting an anti-resorptive effect [39,40].

The effect of BCAA-enriched short peptides in human BMSCs has also been observed [41]. M Huttunen et al. discovered that using 50 μM isoleucine-proline-proline (IPP) could increase BMSC proliferation. However, in contrast, 50 μM leucine-proline-proline (LPP) or 50 μM valine-proline-proline (VPP) showed a very modest influence on osteoblastic gene expression. In the follow-up experiment, IPP was shown to up-regulate Parathyroid Hormone-Related Peptide (PTHrP), BMP-5, and cAMP response element binding protein-5 (CREB-5), and down-regulate Vitamin D receptor (VDR) and caspase-8, which reflects increasing osteoblast proliferation and differentiation.

In 1957, Wasserman et al. observed the effect of lysine on calcium absorption in rat and chicken [42]. In primary osteoblasts, obtained from newborn SD rats, adding lysine, threonine, methionine, or tryptophan can enhance cell growth and ALP activity, as well as collagen synthesis at lower degrees, causing an increase in bone formation [43]. Age-induced bone loss has been associated with the increased generation of reactive oxygen species (ROS), which oxidize the surrounding proteins and enzymes to reduce their normal function. In mouse BMSCs cultured in osteogenic media, oxidized L-tryptophan enhanced cell proliferation and differentiation by enhancing the runt-related transcription factor 2 (Runx2)-osteocalcin and ALP expression. Oxidized L-tryptophan also increased extracellular-signal-regulated kinase (ERK) phosphorylation, which increased cell survival and proliferation [44].

M Refaey et al. discovered that a 50 μM phenylalanine-tryptophan combination could decrease the early and late markers of osteoclasts in cell culture, including the RNA levels of the vitronectin receptor, cathepsin K, indicating that this may inhibit osteoclast differentiation [45]. However, in the in vitro resorption experiment, both phenylalanine and tryptophan increased the resorption pits, which is contradictory to the RNA result. The author guessed that maybe phenylalanine and tryptophan mainly regulate the attachment of osteoclasts. Later, M Refaey continued the research of tryptophan in bone metabolism. In 2017, M Refaey found that kynurenine, which is an oxidation product of tryptophan, accumulated with aging [46]. Subsequent research has demonstrated that kynurenine can induce osteoclast activity by inducing the receptor activator of the NF-κB ligand (RANKL) pathway, and increase the differentiation of adipocytes from BMSCs by inhibiting Hdac3 signaling (Figure 3).

## 5. Relationship between Body EAAs Levels and Bone Quality in Human Research

The role of body EAA content on bone health is inconsistent across various studies. Using Mendelian randomization analysis, Z Cui et al. revealed that isoleucine and valine levels were positively associated with total BMD in European adults [47]. In middle-aged men (31–66 years) with idiopathic osteoporosis, most serum EAAs were not different from that of age-matched controls, except for a significant increase in the threonine level. Erythrocyte amino acids have also been tested and revealed a significant decrease in lysine, phenylalanine, and tryptophan levels in idiopathic osteoporosis patients [48].

A study in China found that in older community-dwelling adults of 1424 men and 1573 women with a mean age of 72 years, serum phenylalanine, tryptophan, methionine, valine, leucine, and isoleucine were significantly lower in osteoporosis subjects [49] (Table 1). These studies suggest that in old people with osteoporosis, the serum EAA levels are usually down-regulated. Most of the studies revealed that EAA levels are positively related with bone conditions, with only Pernow’s study in Sweden showing threonine negatively changing with bone quality [48]. This might be because a very small sample number was used in this study, as the 20 participants, with a large age range from 31 to 66 years old, resulted in a big age variation; thus, this conclusion is not as strong as others.

## 6. Supplementation of EAAs on Bone Quality in Aging People

In 1120 postmenopausal females aged over 50 years, the food source of amino acids was investigated and it was found that leucine and lysine from vegetables are related to a significantly less prevalence of osteoporosis ratios [50]. In addition, monozygotic twins with an intake of leucine and lysine were associated with higher BMD at the spine and forearm, notably, leucine has the strongest association [50]. As a BCAA, there are a lot of studies that have investigated the effect of leucine on bone health, based on the results of this monozygotic twin study.

In a double-blind study of 380 participants older than 65 years, a leucine-enriched whey protein supplement (with a daily supplement of 6 g leucine) for 13 weeks improved the BMD in a small but significant range [51]. C Lin et al. obtained a similar conclusion in another open-label, parallel-group study. Twenty-eight people received a vitamin D- and leucine-enriched whey protein supplement for 12 weeks, and achieved a significant improvement in walking speed in the age 65–74 group [52]. However, other studies have shown neither a benefit nor a harmful effect of leucine supplementation on bone health. In a Japanese study, the aging people were divided into two groups, one taking 0.8 g citrulline and 1.6 g leucine twice daily for 20 weeks, and another group taking a placebo. Both groups were accompanied by exercise, and the results showed no difference in BMD and bone area in the two groups [53]. Another study took place in Northwest England, giving leucine-enriched whey protein drinks to aged 60–90 people for 16 weeks, which also did not show any difference in bone mass between groups [54] (Table 2). The difference in the leucine supplementation effects in Table 2 might be caused by the differences in the sample number, leucine content in the supplement, and the measured index of bone. A small group number (13–30) is not as reliable as hundreds or thousands of sample quantities, and BMD, bone mass, and walking speed do not always change together. In a study in Taipei, leucine was supplied together with vitamin D, which may better improve the absorption of calcium to enhance walking ability [52].

## 7. Conclusions

EAAs are important in aging as a nutritional source and as a signaling regulator. In aging people with bone loss, serum EAA levels are not always consistent in different studies; thus, there is not a well-recognized conclusion about the correlation between serum EAAs and bone volume. In people supplied with extra EAAs, most of the studies showed that EAAs are beneficial for BMD; although, some also showed no effect. The inconsistency of these results is mainly because of the small number of people in the testing groups, and the different nutrients supplied with EAAs at the same time. Therefore, the studies with larger sample numbers are more reliable, which illustrated that EAAs improved BMD. EAAs regulated bone health by increasing the osteogenic differentiation of BMSCs/osteoblasts and by inducing the RANKL-regulated osteoclast pathway (Figure 3).

The function of EAAs in aging bone will be more extensively studied in the future, to better understand it as a metabolic organ. Currently, there are many shortfalls in the human studies, such as the small group of samples, the variations in age, no detection of the absorption rate after amino acid supplementation, etc. In animal and cell studies, some conclusions concerning the functions of EAA-related pathways are conflicted with others. With more studies likely in the future, the correlation between specific EAAs and aging bone metabolism will become clearer.

## Figures and Tables

**Figure 1 ijms-23-11281-f001:**
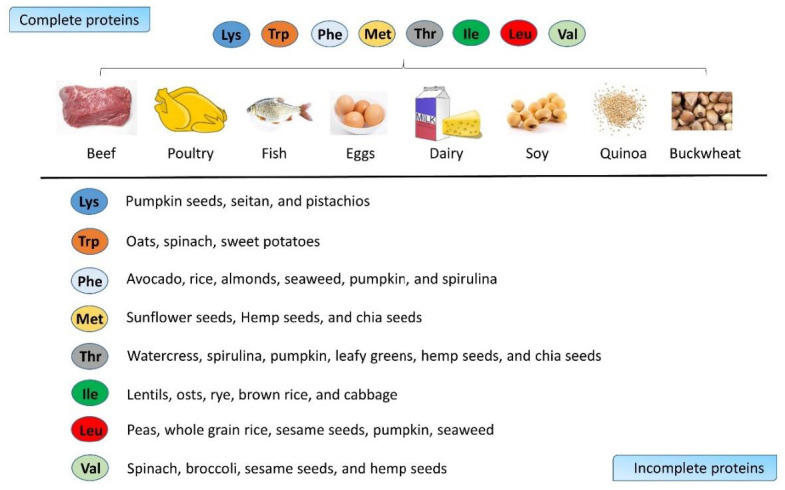
Sources of EAAs from complete proteins and incomplete proteins.

**Figure 2 ijms-23-11281-f002:**
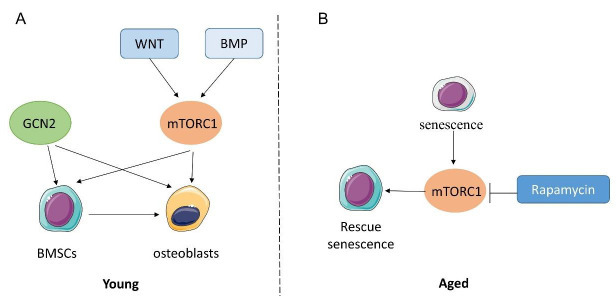
GCN2 and mTORC1 signaling in BMSC-osteoblast lineage cells in vitro at young (**A**) and aged (**B**) stage.

**Figure 3 ijms-23-11281-f003:**
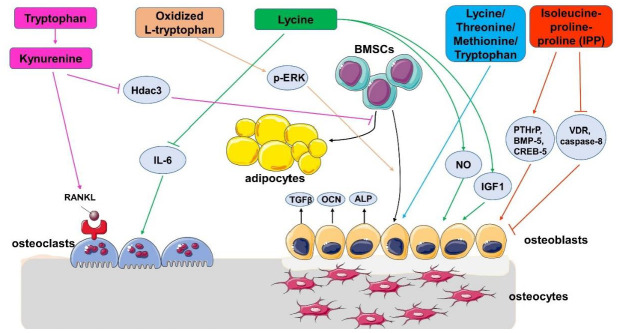
Potential mechanisms of action of essential amino acids in bone metabolism.

**Table 1 ijms-23-11281-t001:** EAA content in aging people with osteoporosis.

Participants Number	Participants Background	Age	Tissue	EAAs	Relationship between EAA Changes and Bone Condition	Publication Year	Reference
502,639	European	37–70	Serum	Isoleucine, valine	Positively associated with total BMD	2021	[47]
20–22	Sweden	31–66 years	Serum	Threonine	Significant increase in people with osteoporosis	2010	[48]
20–22	Sweden	31–66 years	Erythrocyte	Lysine, phenylalanine, and tryptophan	Significant decrease in people with osteoporosis	2010	[48]
1424 men and 1573 women	Chinese	mean age 72	Serum	Phenylalanine, tryptophan, methionine, valine leucine, and isoleucine	Significantly lower in osteoporosis subjects	2019	[49]

**Table 2 ijms-23-11281-t002:** Impact of EAA supplementation in aging bone.

Participants Number in Each Group	Participant Background	Age	EAAs Supplement	EAA Treatment Time	Changes after EAA Supplementation	Publication Year	Reference
184–196	6 European countries: Belgium, Germany, Ireland, Italy, Sweden, and the UK	Older than 65 years	6 g of leucine with 40 g of whey protein daily	13 weeks	BMD increasing in a small but significant range	2019	[51]
28	Taipei	Older than 65	1.2 g leucine enriched whey protein supplement	12 weeks	Significant improvement in walking speed in age 65–74 group	2021	[52]
13	Japanese	65–80 years	1.6 g leucine twice daily	20 weeks	No effect on BMD and bone area	2021	[53]
22–31	Northwest England	60–90 years, mean age 68.73 ± 5.80 years	1.50 g/kg BW/day whey protein, plus 0.03 g/kg BW leucine	16 weeks	No effect on bone mass	2021	[54]

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
