# Peer review of "Role of Essential Amino Acids in Age-Induced Bone Loss"

_ijms, 2022, doi:10.3390/ijms231911281_

Round 1
Reviewer 1 Report
The review should be more careful edited in English. For example pg1/row 36, or should better explaine the abbreviations, such as at pg3/r82 – M+4, what does it mean?
In the paragraph 6. Supplementation with SCAA...you should comment on the studies outcome presented in table 2, on the bases of the dosage and duration of supplementation.
Author Response
The review should be more careful edited in English. For example, pg1/row 36, or should better explained the abbreviations, such as at pg3/r82 – M+4, what does it mean?
Response:
Thank you for pointing out. We changed the grammar mistake in page1/row36.
In page3, we added the explanation of M+4 at line 83-87 as follows:
Mass spectrometry technique is used to detect the metabolic tracking from stable isotope tracers. 13C is the predominant isotope used because of its universal distribution in almost every cellular organic molecule. In the metabolic substrates analysis, M+0 is unlabeled molecule, while M+1, M+2, M+3, etc. are different numbers of carbon labelled with 13C.
In the paragraph 6. Supplementation with SCAA...you should comment on the studies outcome presented in table 2, on the bases of the dosage and duration of supplementation.
Response:
Thank you for the advice. We added the comment about the studies at ling 214-219 as follows:
The difference of leucine supplement effects in Table 2 might be caused by the differences in sample number, content in supplement and measured index of bone. Small group number (13-30) is not as reliable as hundreds or thousands of sample quantities, and BMD, bone mass and walking speed are not always changed together. In the study at Taipei, leucine was supplied with vitamin D together, which may better improve the absorption of calcium to enhance walking ability.

Reviewer 2 Report
This is an interesting article. Some minor points are suggested below.
1. Line 97, please avoid abbreviation when mentioning WNT in the first time.
2. Avoid unnecessary capital letters, such as but not limited to Nitric Oxide (line 130), alkaline phosphatase (line 131).
3. Cite more pertinent MDPI publications.
4. Line 138. Correct the misspelled "leulin" .
5. Table 1. Avoid mixing the columns. The EAAs changes with skeletal condition should avoid mixing together. Besides, "participants" , "Publication year": try to unify the use of capitalized letter.
6. Substantial grammatical minor errors should be corrected. Such as, but not limited to , " have not reach" (line 171), " base on" (line 190).
Reviewer 3 Report
The manuscript "Role of Essential Amino Acids in Aging-induced Bone Loss" addresses an important issue: Aging induced osteoporosis and the role of essential amino acids (EAAs) as an energy source and a molecular pathway modular in bone. In this article, the authors discuss the contribution of EAAs in aging bone as energy provider and as molecular signal modulator both in vitro and in vivo.
The objectives were clearly stated and explained in the manuscript, and the experimental strategy was appropriate to gather the experimental information from which the conclusions were drawn. The manuscript is well written and has good organization. The authors have done a great job on analyzing the experimental data and on discussing the results, considering always different alternative explanations/considerations for interpreting the results.
Some major points deserve careful attention:
1. The authors should discuss more extensively on why some studies showed EAAs are beneficial for the BMD and discuss the reasons behind other studies not indicating these benefits.
2. In the Introduction section the authors may address the effect of EAAs in other physiological processes better understood such as muscular growth more in depth as this may help clarifying some of the effects seen in other physiological processes such as those related to bone homeostasis.
3. In the Conclusions section the authors should address more thoroughly the dissonances detected during the research and why do they originate
Minor points:
1. In vivo and in vitro as well as other Latin terms should be italicized.
2. Caption in Figure 1 should be extended to become more comprehensive.
3. Line 121 “agd” should be changed to “aged”.
Round 2
Reviewer 2 Report
This work in essence is very interesting. I have only raised some minor suggestions in the latest round of revision, and my suggestions have been appropriatedly addressed by the authors. This work is suitable for publication now.
Reviewer 3 Report
After reviewing the revised version of the manuscript and the responses of the authors that answered all the presented questions my decision would be to accept the manuscript.